# Review of Kalah Game Research and the Proposition of a Novel Heuristic–Deterministic Algorithm Compared to Tree-Search Solutions and Human Decision-Making

**Libor Pekař [1],\*** , **Radek Matušů [2]** , **Jiří Andrla [1]** and **Martina Litschmannová [3]**

1   Department of Automation and Control Engineering, Faculty of Applied Informatics,
    Tomas Bata University in Zlín, 76001 Zlín, Czech Republic; jiriandrla@seznam.cz
2   Centre for Security, Information and Advanced Technologies (CEBIA—Tech), Faculty of Applied Informatics,
    Tomas Bata University in Zlín, 76001 Zlín, Czech Republic; rmatusu@utb.cz
3   Department of Applied Mathematics, Faculty of Electrical Engineering and Computer Science,
    VSB—Technical University of Ostrava, 70800 Ostrava-Poruba, Czech Republic;
    martina.litschmannova@vsb.cz
\*   Correspondence: pekar@utb.cz

**Abstract:** The Kalah game represents the most popular version of probably the oldest board game ever—the Mancala game. From this viewpoint, the art of playing Kalah can contribute to cultural heritage. This paper primarily focuses on a review of Kalah history and on a survey of research made so far for solving and analyzing the Kalah game (and some other related Mancala games). This review concludes that even if strong in-depth tree-search solutions for some types of the game were already published, it is still reasonable to develop less time-consumptive and computationally-demanding playing algorithms and their strategies Therefore, the paper also presents an original heuristic algorithm based on particular deterministic strategies arising from the analysis of the game rules. Standard and modified mini–max tree-search algorithms are introduced as well. A simple C++ application with Qt framework is developed to perform the algorithm verification and comparative experiments. Two sets of benchmark tests are made; namely, a tournament where a mid–experienced amateur human player competes with the three algorithms is introduced first. Then, a round-robin tournament of all the algorithms is presented. It can be deduced that the proposed heuristic algorithm has comparable success to the human player and to low-depth tree-search solutions. Moreover, multiple-case experiments proved that the opening move has a decisive impact on winning or losing. Namely, if the computer plays first, the human opponent cannot beat it. Contrariwise, if it starts to play second, using the heuristic algorithm, it nearly always loses.

**Keywords:** C++ implementation; cultural heritage; decision-making; game playing; heuristic strategy; human-computer interaction; Kalah; Mancala; survey; two-player games

## 1. Introduction

The art and ability of playing games are widely considered as a kind of human intelligence. Game playing represents the most suitable environment for studying complicated problem-solving [1]. This paradigm exists because games are perceived as models of real-life decision-making, social behavior [2,3], conflict situations [4], knowledge acquisition [5,6], economic prosperity [7], or even climate change perspectives [8] throughout various nations and cultures [9]. The rules of a particular game represent a counterpart of the corresponding social norms [10], cultural practice, and given

legislative. Nowadays, terms such as "good games" [11], "serious games" [12], "gameful design" [13] or "gamification" [14] (expressing the general concept of the use of games and their elements for a goal beyond mere entertainment in a non-gaming context) have become more and more popular, especially in software-based games. Educational objectives are the main reason for designing these concepts.

Board games represent a subset of the so-called adversarial games in which every single player tries to beat one or more other players, usually without a the possibility of cooperation or any kind of trade-off. They have fascinated people as mirrors of intelligence, skill, cunning, and wisdom throughout various nations and cultures for thousands of years [9,10,15,16]. Indeed, Roberts, Arth, and Bush [17,18], in their pioneering work, delineated a three-category classification for games. These are namely: physical skill, strategy, and chance. Hence, board games can be covered by the latter two classes and represent a considerable portion of all games. A widespread opinion is that most of the board games originate from ancient Egyptians for the Predynastic period about 5000–3000 BCE [19] and they were later distributed by way of caravan merchants to the Arabian Peninsula (probably to the great Nabataean cities [20]) and then to the entire world.

The family of Mancala games is a famous, favorite, and one of the oldest representatives of board games ever [21]. The term "mancala" comes from the Arab word "nagala" for "move" [22]. According to de Voogt [23], this group of board games commonly consists of two or four rows of cup-shaped holes (pits) in which a proportionate number of counters (seeds, stones, shells, etc.) is placed. Usually, two players take turns spreading seeds one by one in consecutive holes around the rows of holes, which is called the sawing. The goal of the game is to capture as many seeds as possible in most cases. There are various types of boards and rules. For instance, games of Wari (the most widespread Mancala game), Owari, Ayo, and Soro known from West Africa and the Caribbean, Bao (or, Boa), and Omweso played in East Africa, Vai Lung Thlan played in India, etc. More recent or modern versions of the game are Awari [24] or Kalah. Nowadays, more than 800 names for various Mancala games are known worldwide [25,26]. Different types and rules of Mancala are closely related to the way of thinking of people and create a form of communication. For instance, the player can estimate the skills and mind of the opponent; and the opponent may show the spectators that they are a weak player [22]. Due to their simple rules and variety of game positions, Mancala games have become a popular tool for practicing mental and mathematical skills at schools [27,28], besides, e.g., the habitual chess game.

As mentioned above, Kalah represents a relatively modern Mancala variant that was introduced by William Julius Champion Jr. in the 1940s [25,26]. It became highly popular, especially in the western part of the world [9,29]. It is played with a board of two rows of six pits. A particular player owns each row that is equipped with two stores (end-zones, houses, or kalahas) with a special meaning, see Figure 1 [30]. One player owns the south side, whereas another player sits in the north one. There are four seeds (counters) inside each pit at the beginning of the game. Each player makes moves (sawing) to collect as many seeds as possible within the given rules. It is possible to gain an additional move or to capture the opponent's seeds (that are placed to the player's store) in some cases. However, the game position cannot repeat, and the final move can be made such that all opponent's seeds are eventually captured (note that the rules are described in Section 3 in more detail). The same board is used for Wari, Oware, and Aware, while their rules differ. For instance, the sawing is possible only for some of the pits, seeds from pits of different features can be captured, multiple captures are possible, a game position can be repeated, etc. [26].

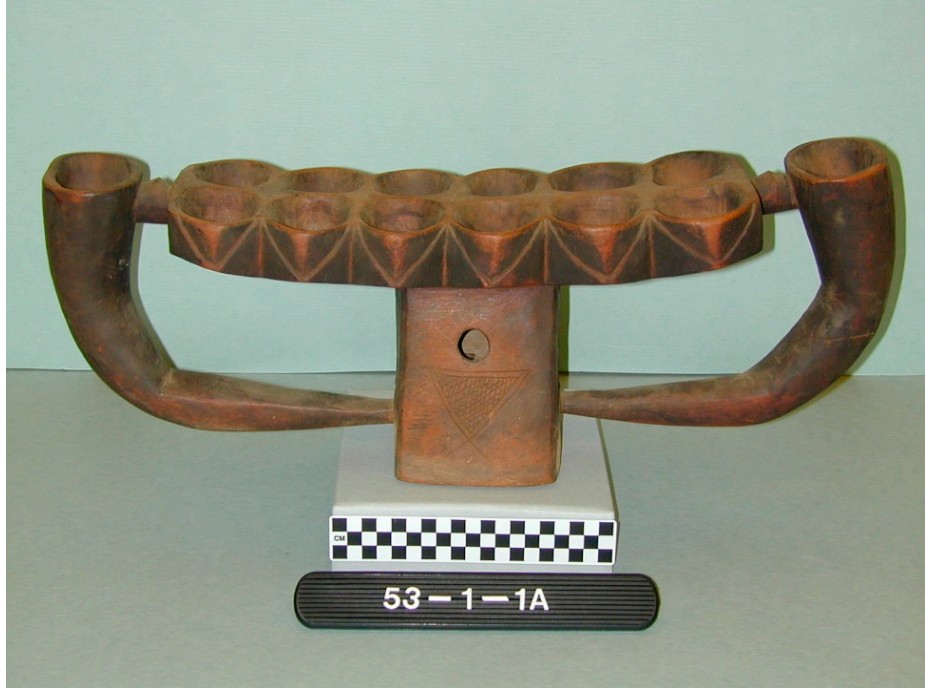

**Figure 1.** Historical Mancala game board with two rows of six pits and a store usable for the Kalah game [30].

Various studies have been made to play Kalah, see, e.g., [1,24,26,31–36]; however, the game was utterly solved for its many variants by Irving et al. [29] for the first time. A very high decision complexity of Kalah [37,38] prevents the game from being solved earlier. However, this solution was made by a brute force via advanced game-tree search. Although such a technique is useful in terms of the complete game solution, its implementation might be a little demanding. Therefore, it is reasonable to develop and apply procedures that reduce the computational costs [38] and still provide a very good chance for a player to win (or draw at least). This goal can be met by the use of knowledge-based or heuristic methods [26,32,39,40], which require the analysis of game rules and position variants followed by reasonable suggestions on how to move. A combination of these methods with brute-force (e.g., game-tree search) solving methods may lead to the reduction of the so-called solution size [38] and yield an outstanding performance compared to some standard methods [37]. On the contrary, it should be noted that rapid technology development enables us to apply more and more complex and memory-demanding hardware solutions nowadays.

This paper is primarily aimed at a comprehensive survey on solution methods and further analyses made by researches and practitioners. It provides the reader with sufficient insight into the ideas of how these issues have been tackled so far and it serves as a stepping stone for further problem investigation, which may be useful for other involved researchers and programmers. One can consider that as a tribute to the cultural heritage. Mainly, it is attractive and desirable for understanding the human way of thinking and decision-making. This task, however, cannot be solved by brute-force approaches.

Hence, we further attempt to support this endeavor by the proposition of an original computationally simple heuristic algorithm for playing the Kalah by a computer, as the second paper goal. The method is based on selected heuristics that are represented by ordered deterministic moving strategies. The heuristics are based on the analysis of rules of the standard Kalah game version.

The efficiency of the proposed winning strategies is evaluated by the comparison with the basic and a modified mini–max game-tree-search algorithm [26,41], with a specific conditioning and evaluation function, of the depths 1, 2, 4, and 6 to get the computer response in a reasonable time. Moreover, a tournament against a medium-experienced amateur human opponent is performed. The obtained results are evaluated by some statistical charts displaying the success rate. These two

bunches of experimental results prove a very good performance and real-life applicability of the presented heuristic algorithm.

Some theoretical results refer to the winning advantage of the first (opening) move, i.e., that of the first player, for many game variants, see, e.g., [29,32,36–38]. We attempt to verify this assumption in this paper as well.

A very simple application programmed in C++ language equipped with the Qt framework is developed, the Graphic User Interface (GUI) of which constitutes a kind of the human–machine interface to perform gaming experiments. A concise description of its functionality is provided as well.

The rest of the paper is organized as follows. A concise overview of the Mancala games family history and, especially, of research pursued and published so far, and state-of-the-art survey for Kalah are given to the reader in Section 2. In Section 3, rules of the standard Kalah game are introduced and analyzed to determine suitable heuristics (strategies). The identified deterministic heuristics are weighted against some suggestions found in the literature. In conclusion, the eventual algorithm is concisely presented in Section 4. Section 5 introduces two versions of the game-tree-search mini–max algorithm that serves as a benchmark. First, the standard mini–max is described; second, its semi-heuristic modification, specific to Kalah, is developed. The developed GUI of the simple application is concisely given to the reader in Section 6. The evaluation of the proposed method is performed in Section 7 in the form of two tournaments with the human opponent and against the two tree-search algorithms. Finally, this paper and its findings are concluded in Section 8.

## 2. Mancala/Kalah Games History and Research Background

This section is aimed at a concise introduction to the history of Mancala games and one of their recent forms—Kalah. As the primary goal of the section, selected significant research results on playing and solving Kalah or some rules-related Mancala versions are reviewed, which gives rise to the collateral motivation of this paper.

### 2.1. History of the Game

Mancala (also known as "count-and-capture") board games cover a family of games played usually by two players on boards with two to four rows of pits [23]. Mancala games have been associated with the Arabs and the Ottoman Empire [20] so far; however, the origin of these games goes back thousands of years, it is still unclear and poses a sort of mystery. Crist et al. [19] place the origins of Mancala to ancient Egypt, based on a thorough literature overview. For instance, they refer to the discovery of several examples of rows of pits cut at the entrance of the temple of Karnak and at the Luxor temple from Ptolemaic times (~200–300 BCE). In Dendera, a board with two rows of six pits and also that of two rows of five pits equipped with two additional holes at each far end were found. They state that several types of games with hollowed or divided boards and counters were played by Egyptians as early as 4000 BCE. On the contrary, Rovaris [26] considers that Mancala was spread through the Bantu expansion; however, the Bantu nation lives south of the equator.

Conclusions on the distribution of these games in history are complicated by the relative absence of such playing practices in the historical record [20]. Many board games may be confused due to their variants. For instance, the so-called tab games resemble four-row Mancala games. Moreover, various cultures and nations may share the same archeological site with similar yet different game boards and rules. Charpentier et al. [20] refer to a board of two rows with five holes discovered in excavations of a Roman fort in Egypt and Palmyra (Syria). Whereas Mulvin and Sidebotham [42] believe that is associated with a Mancala game by Romans, Schaedler [43] means that it was used to a game of five lines, not associated with Mancala. It was the only type of board in Palmyra from the Roman Empire; however, Arabs and Ottomans used different configurations in this site later [44]. In any case, it is commonly believed that this group of games was spread to the Arabian Peninsula, the Asia Minor, many parts of Africa, and to the ancient world by Ottomans during the expansion of Islam. The credit for their expansion to Western Europe also belongs to the Romans [45]. These games could also have

been distributed by way of caravan merchants to the great Nabataean cities [20]. They were abandoned in some regions after the departure of the Ottomans; however, they are played throughout almost all continents nowadays [26]. Mancala games are a favorite, especially in Africa, where it is not rare that different game rules for the same name or different names for the same game within a relatively small area are used. However, these games show only minor changes throughout many borders, cultures, and years, which proves their high abstract characteristics, the reproducibility, and transferability, independent from a particular cultural influence [46,47].

Kalah represents a commercial variant of Mancala games invented by a former student of Yale University in the 1940s of the last century and consequently marketed by his The Kalah Game Company in the 1950s [29]. The game appeared on computers for the first time already in 1960 [48]. Nowadays, it is the most popular version of Mancala in the United States and Western Europe. Due to the cultural invariance of board games, the high popularity of Kalah, and its simple rules, this game is an excellent object of study and a benchmark of game-solving and artificial intelligence methods. A chronological/topical overview of research on Kalah follows. Some necessary notions are introduced.

*2.2. Mancala Research Survey*

As early as in 1964, Russell [49,50] presented two studies on playing Kalah on the computer. A partial self-learning computer program to demonstrate the game was developed by Bell in 1968 [48]. Slagle and Dixon [51] programmed several board-games, including Kalah, to illustrate and test their M&N algorithm two years later. From that time, no relevant result has been published until the end of the millennium. However, some end-game databases were built for Awari [24], the rules of which are close to Kalah. The end-game database indicates whether a player wins, losses, or draws from any possible game position for any allowed move. The size of such a database depends on all possible positions expressing a particular seed distribution over the pits of the board. If *m* is the number of pits per side (except for the stores) and *N* stands for the total number of initial seeds, then the total number *p*(*m*,*N*) of such positions reads [26,34].

$$p(m, N) = 2 \binom{2m + N + 1}{N},$$
(1)

where the information about which player moves next is included. Since seeds captured into stores cannot re-enter the game, the number of active seeds decreases during the play; hence, the number of possible positions decreases as well. Therefore, Irving et al. [29] determined the formula for the set size *s*(*m*,*N*$_a$) of all configurations with the number $N_a$ of active seeds, regardless who is playing next, as follows

$$s(m, N_a) = \binom{2m + N_a}{2m} - 2 \binom{m + N_a}{m} + 1,$$
(2)

where configurations with no seeds on one of the board sizes are left out. The values of both *p*(*m*,*N*) and *s*(*m*,*N*$_a$) increase rapidly with respect to the parameter values. The standard game with *m* = 6, *N* = 48 (i.e., four seeds per pit) yields $p = 1.313 \times 10^{13}$ and, for instance, if *N* = 24, then $s = 1.25 \times 10^9$. However, only about 5% of all these positions can appear in a real game [29,34]. Romein and Bal [35] solved Awari for the perfect play of both players. They used a large computing cluster and implemented a parallel search algorithm that determined the results for almost $9 \times 10^9$ positions in a 178-gigabyte database.

Regarding the solution of Kalah, the work of Irving et al. [29] represents the fundamental contribution. The full-game databases were constructed (via the full game-tree search and the backward evaluation) for smaller instances of Kalah, up to *m* = 4, *n* = 3 where *n* is the number of seeds per pit at the very beginning of the game. It represents the so-called strong solution, i.e., a strategy is known to achieve the game-theoretic value of all (not only the initial) possible board positions [31]. This value means the outcome of the game when both the players behave optimally (the so-called perfect play), i.e., whether the game is won, drawn, or lost for the first player [37]. A so-called

weak solution for $m = 4$ to 6, $n = 1$ to 6 (except for $m = n = 6$) in the form of an advanced game-tree search was found in [29] as well. This searching algorithm applies MTD($f$) [52] (i.e., an improved alpha-beta pruning) with iterative deepening, including futility pruning [53], transposition tables, a move-ordering, and an end-game database (with $m = 6$, $N_a \leq 30$). Note that the weak solution means that a strategy is known to achieve the game-theoretic value of the game from the initial position [31,54]. Smith [36] in his survey on the use of dynamic programming for board games referred to the advantages and disadvantages of the above-introduced works [29,35]. A detailed analysis of the game with the combinations $m = 1$ with $1 \leq n \leq 300$ and $1 \leq m \leq 4$ with $n = 1$, and the eventual numerical observations were reported in [32]. Pok and Tay [55] proved that every game starting with $m = 1$ and 11 . . . 11 (in ternary notation) seeds for each player yields a loss for the first player. Moreover, they proposed a modification of Kalah.

Results of a brute-force computer programming endeavor by M. Rawlings are referenced in [56,57]. It is reported therein that end-game databases for $m = 6$, $n = 4$ to 6 were computed within the years 2015 and 2018. In contrast to [29], each of the initial moves with the further perfect play was analyzed. Besides, they constructed end-game databases, including the perfect play results for $m = 6$, $N_a$ up to 35; i.e., Equation (2) gives $s = 52.24 \times 10^9$ possible positions.

An excellent survey by Herik et al. [37] analyzed several two-player board games in terms of used brute-force and knowledge-based methods. Regarding the latter ones, the authors discussed two algorithms. First, the threat-space search [31] that investigates whether a win can be forced by a sequence of threats (to which the opponent has only a limited set of replies at any time). Second, the proof-number search [58] where the cost function is given by the minimum number of nodes that have to be expanded to prove the goal. In this survey, it was also questioned whether the knowledge obtained from solved games be translated into rules and strategies which human beings can assimilate and whether these rules are ad-hoc or generic for various games and their complexity. The authors concluded that the answers are dependent on the capacity of the human brain, and the rules are ad-hoc and hardly intelligible to human experts.

Tree-search algorithms have become popular in the reign of board games. In contrast to brute-force full-game databases, they provide faster searching for a suitable move strategy and usually adopt some artificial intelligence (AI) or heuristic approaches [59]. Usually, it is supposed that both the players behave optimally—in such a case, the standard mini–max tree-searching algorithm is used to get the evaluation function maximum value for one player and the minimum for the opponent in every single depth of the game tree. In [60], an overview of popular and effective enhancements for board game using Monte Carlo tree search (MCTS) agents is given. The MCTS is based on a randomized exploration of the search space [61]. Ramanujan [62] studied, inter alia, the efficiency of a family of mini–max methods, and the Upper Confidence bounds applied to Trees (UCT) [63], an advanced sampling-based planning approach, used to general Mancala games. They claimed that both the UCT and mini–max produce players that are competitive with nearly no enhancements to the basic algorithms in Mancala. A combination of the mini–max game-tree search and heuristics was proposed in [38]. Even if used for a Domineering board when perfect solving (defined as solving without any search), Uiterwijk [40,64] discussed the advantage of a heuristic that might be very useful in Kalah as well—safe moves. A safe move cannot be prevented by the opponent. In his bachelor's thesis, Berkman [65] summarized and benchmarked several advanced mini–max techniques for Toguz kumalak (a Mancala game close to Kalah), such as alpha–beta pruning, iterative deepening, hash functions, the MCTS, and transpositions tables. An excellent, thorough, and extensive analysis and benchmark of AI game-tree algorithms for several board games (including Kalah) were provided in [26]. Compared to the preceding referenced source, the greedy algorithm and advanced heuristic mini–max [38] (that uses a more refined heuristic function) were applied in addition. The results proved the advantage of the heuristic approach. Perlinski [66] proposed a faster search by using the so-called partial board tree search that involves dividing the game board into smaller parts. In [56],

several Kalah aspects and tactics by Chris Bandy are referred. Note that some of the above-mentioned heuristic approaches are introduced in more detail in Section 3.2.

Akinyemi et al. [67] proposed a hybrid combination of the mini–max and a machine-learning-based refinement applying the perceptron algorithm [68] for the Ayo/Awari game. They used "priority moves" that give the player a better advantage; however, their meaning remains unclear due to a poorly detailed description. Oon and Lim [69] used the co-evolution of artificial neural networks on Kalah, where the input layer represented the current board position, and each input was an integer equal to the number of seeds in the corresponding pit. The so-called opponent-model search [70] for board games (namely, the Bao game was used as a testbed) was investigated in detail in [71]. This game-tree search algorithm uses a player's hypothesized model of the opponent moves in order to exploit weak points in their strategy; it assumes that the opponent uses the mini–max algorithm, and their evaluation function is worse than the player's one. Birrell [72] investigated and discussed the results of applying reinforcement learning to train an agent to play an (unspecified) Mancala game with the rules almost identical to Kalah. Here, the agent saves the board state to a stack at the start of each turn. At the end of the game, it goes through each board state and adjusts the weight of the move according to the agent's success. Training sets begin with an untrained agent that chooses each move randomly. Yet, only one-step ahead moves were evaluated in [72].

Some scholars discussed the advantage of the first turn yielding the winning end of the game. Donkers et al. [73] proved the winning opening for the Dakon game, the rules of which are close to Kalah. Irving et al. [29] computed that higher values of $m$ and $n$ ($m + n > 6$) yield the advantage of the first player when perfect playing; however, this rule has exceptions (e.g., $m = 3$, $n = 6$ results in a loss for the first player). They particularly proved a win by 10 for the first player with a perfect play for the standard game ($m = 6$, $n = 4$). Carstensen [74] filled in the gap of [29] and proved the first-player win for $m = n = 6$; this win is by 2 [56]. In [32], a summarizing table of the first player's wins/draws/losses for $2 \leq m \leq 6$ and $1 \leq n \leq 10$ (except for some higher values of the sum $m + n$) is displayed for a specific simple heuristic strategy (see Section 3.2). Rovaris [26] referred to a slight bias towards the first player (48.6% vs. 44.9%) when combating several strategies randomly. They believed that it is because the player going first is the first one able to gain multiple moves. Herik et al. [37] found that there is a clear correlation between the first player's initiative and the necessary effort to solve the game. In [72], it was initially supposed that the first player could win 70% of all games; however, no significant advantage of the first player was eventually approved for a trained agent.

Another problem is to propose a suitable and meaningful position evaluation, i.e., to determine the cost function. Donkers et al. [73] defined three different types of evaluation functions expressing prediction of the game-theoretic value, probability of winning, and profitability of the current position. A cost function in the form of a weighted combination of six heuristics (strategies) of the player and the opponent was proposed in [38]. In [65], the cost function is simply given by the number of stored seeds, the weights of which are set by a genetic algorithm. Note that the use of genetic programming when playing Mancala games was discussed in [75].

Only a few research results were dedicated to a rigorous mathematical analysis of the game based on the combinatorial theory. Broline and Loeb [76] studied a certain end-game correspondence between a Mancala game Ayo (Awale) and the solitaire game Tchoukaillon. In particular, they analyzed the possibility of repeated moves (i.e., when the last seed ends in the player's kalaha). There exist schemes that enable to clear a particular number of pits by these moves there. It was proved that the minimum number of seeds required for such a clearance scheme for $m$ pits is asymptotically approximated by $\pi^2/m$. An extension of a novel method for constructing move vectors in Tchoukaillon to Mancala games was proposed in [77]. Musesti et al. [78] derived the lower and upper bounds required to reach periodicity in Oware. However, these results are purely theoretical and hardly applicable when practicing the game.

Kalah has relatively low state-space and game-tree complexity, these are $10^{13}$ and $10^{18}$, respectively [54]. For instance, Chess has $10^{43}$ and $10^{123}$, respectively. Besides these two complexity

measures, an alternative concept (based on the games solved until the date) to determine the solvability of two-player games was introduced in [38], see Figure 2. The authors pointed out that, e.g., Qubic game was solved in 1980 while Awari in 2003—but the former one has the lower state-space and the game-tree complexity than the latter one. They introduced the notion of the solution size that refers to the number of positions for which the optimal move or strategy must be stored in a certificate proving the solvability of a game. This notion extends the decision complexity expressing the number of decisions that are required to store a solution [31]. It holds that the solution size is smaller or equal to the decision complexity. The Awari game has a very high solution size of almost $9 \times 10^{11}$ [35,38], and it is expected that Kalah has a similar one.

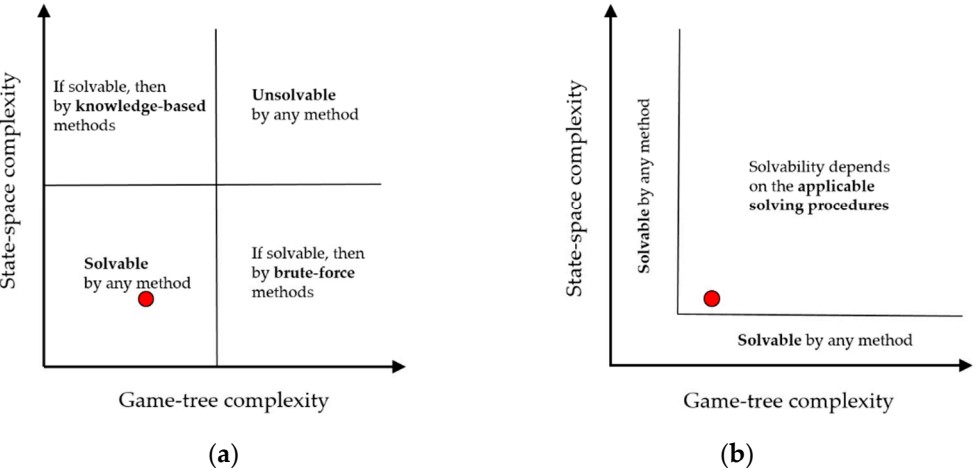

**Figure 2.** The classical (**a**) and an alternative view (**b**) on the game space according to [38]. The red dot indicates an approximate position of Kalah $\{10^{13}, 10^{18}\}$ [54].

Five categories/techniques (retrograde analysis, transposition tables, game-tree search procedures, winning threat-sequence methods, and winning pattern procedures) were discussed to reduce the solution size, and the retrograde analysis [79] (starting at the terminal position and working backward) combined with the end-game databases was identified as a suitable solving procedure for games with large size. However, it represents a brute-force method that is suitable for solving the game, not for real-time playing, as discussed below.

*2.3. Novel Heuristic Method Motivation*

As can be seen from the above-given literature overview, the Kalah game was investigated mostly by brute force or by modifications of the game-tree search mini–max algorithm. The former methods may provide a weak or strong solution by exploring numerous board positions in end-game or full-game databases. However, these solutions are not ideal for real-time game playing. The reason for this claim is twofold: (1) a thorough computation of all or almost all possible positions may yield a slower move-decision computation on the computer side. The computation time is shorter compared to brute-force or game-tree methods, which makes heuristics more appropriate for the game practicing. (2) Such methods do not say anything about the human way of thinking about the moves to be selected. Although it was suggested to solve Kalah by brute-force methods in [37,38] (due to its high decision complexity), this reasoning came from the analysis of a small number of game solutions made until the date. As already mentioned, a weak point of use of the mini–max (from the player's perspective) when practicing the game is that it supposes that the opponent behaves optimally. However, it can hardly be reached by a real human opponent. In such a case, the mini–max algorithm may lead to the evaluation function value much lower than the game-theoretic value. Therefore, it opens a way to the use of heuristic methods and strategies (or other knowledge-based methods) that can be proposed for Kalah ad-hoc [58].

Although these methods may not provide a solution to the game, the examination of the proposed computer heuristics and ad-hoc strategies by playing against humans can serve for their playing behavior investigation and analysis. The first step must be to analyze the game rules and to propose strategies that represent a "good" choice expressed by a move through which, e.g., the player loses nothing or boosts their score. Defensive strategies have to be naturally taken into account as well, which prevents the player from being attacked by the opponent. Moreover, each of the strategies needs to be valued by its weight. If these strategies have a win rate close to the human one, it may simulate our way of thinking and provide new opportunities for further research.

On the contrary, whenever the computer plays much better than the opponent or it is defeated in most cases, the proposed strategies go beyond the human-brain limits or do not represent true human decision-making, respectively. Even though the former task can be an attractive and exciting problem to solve, it remains future research due to its enormous complexity. It is interesting that knowledge-based methods applied to some games with human-understandable rules (e.g., Go-Moku [80]) may not be sufficient to solve the game, which coincides with the latter problem. Hence, the research question is whether a heuristics-based algorithm can achieve a mid-experienced amateur human player in Kalah.

Despite that it has already been solved, Kalah remains an excellent game to analyze all the issues raised above. It has simple rules, a limited range of choice of moves, but the number of possible positions is considerable. Besides, its attractiveness is also given by the fact that it has the branching factor (approx. 4.08 for the classic version of the game [29]) very close to the popular game of checkers (approx. 2.84 [69]), which may yield the portability of the obtained results between games.

Last but not least, understanding playing Kalah—as a representative of the family of Mancala games, dating back a long way—definitely contribute to world heritage.

## 3. Kalah Rules and Their Analysis

Rules of the very classic version of Kalah with $m = 6$ and $n = 4$ are introduced in this section. An analysis of the rules follows, which eventually gives rise to our heuristics and strategies proposed in the further section.

### 3.1. Rules of the Game

The two players sitting at each side of the board control their own pits and the store (kalaha) on their right-hand side. Let the player having the south side be simply called "the player", whereas the second one sitting at the north side be called "the opponent" hereinafter, see Figure 3 for the starting position. In each turn of the game, both the players alternate; however, repeated moves are possible (see below). A turn from the player's perspective is made as follows [26,29,32,55,56]:

1.　One of the non-empty pits is selected, all the seeds are removed from it and sawed to adjacent pits, including the player's kalaha, one by one in the anti-clockwise direction.
2.　When sowing, the opponent's kalaha (on the left-hand player's side) is skipped.
3.　If the last seed of a move is sown in the player's store, the player has an additional move within the turn (i.e., go to item 1), see Figure 4 for an example.
4.　If the last seed of a move is sown in a player's empty pit and the opposite one is non-empty, this seed and all the seeds in the pit on the opponent's side of the board are captured into the player's kalaha. See Figure 5 for an example of such a move.

Seeds captured in both the kalahas do not re-enter the game. Regarding item 3 of the rules, a chain of multiple moves can be achieved. Regarding item 4, it is worth noting that the so-called empty capture is not applied here. This rule means that the catch is made even if the opponent's opposite pit is empty, i.e., the only one player's seed is stored in the kalaha. The game ends when one of the players no longer has seeds in any of their pits. The remaining seeds are captured by their opponent and eventually stored in the kalaha. The player with more seeds in their kalaha is declared as the game-winner.

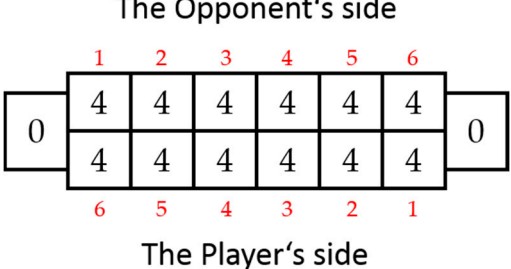

**Figure 3.** The initial board position of the game. Black numbers indicate the number of seeds inside a particular pit (kalaha), whereas red ones mean pit numbers (positions).

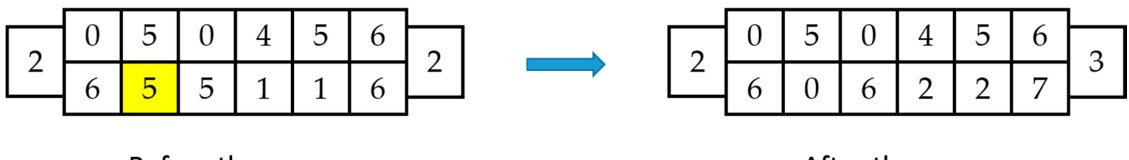

**Figure 4.** A move leading to an additional move. The selected pit to be sowed is highlighted in yellow.

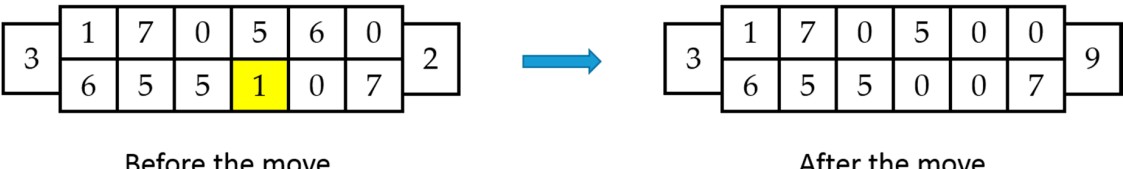

**Figure 5.** A move leading to the capture of the opponent's seeds. The selected pit to be sawed is highlighted in yellow.

Kalah is a monotonically diminishing game, in that a position can never be repeated [69] in a particular game. As a result, the games of Kalah seldom last more than 30 moves.

### 3.2. Analysis of the Rules

In this subsection, Kalah rules are analyzed to propose efficient and straightforward heuristics. A concise review of heuristics used so far is given in addition.

The source [56] refers to several tactics by Chris Bandy where some of them are further subdivided into subtactics. It defines a "stash" as an opponent's pit with a large number of seeds inside and a "target" as the player's empty pit. The basic tactics (from the player's perspective) T1 to T6 are as follows:

**T1:** Make a move resulting in the capturing of an opponent's stash.
**T2:** Make a move that may yield T1 in the next move (or a turn).
**T3:** Remove the threat made by the opponent's move T3.
**T4:** Make a series of moves within a single round.
**T5:** Reach T2 by more than one pit.
**T6:** Reach T2 if the opponent has reached T2 in the last round as well.

It is worth noting that T1 can be achieved within T4. Tactics T5 and T6 can be viewed as special cases of T2.

In [32], a simple strategy was applied. This strategy attempts to move the seed(s) from the pit in which the last seed will land in the kalaha whenever possible; otherwise, the player attempts to capture opponent's seeds. It means that tactics T4 is preferred. If such a move is impossible, then T1 (or T2) is attempted. However, this strategy gives a win/loss map for the first player for various numbers of

pits and seeds inside that is far from the game-theoretic values [29]. Thus, it is not suited to be used standing alone. Notably, this heuristics ignores any kind of defense.

Divilly et al. [39] proposed an original evaluation (cost) function of the evolved player within the mini–max game-tree search as

$$f = \sum_{i=1}^{5} w_i h_i - w_6 h_6 \tag{3}$$

where $w_1 = 0.199$, $w_2 = 0.190$, $w_3 = 0.371$, $w_4 = 1$, $w_5 = 0.419$, and $w_6 = 0.566$ are weights found by a genetic algorithm and $h_1$ to $h_6$ mean values of heuristics (i.e., goals to reach) H1 to H6 formulated as follows:

**H1:** Collect as many seeds in one pit as possible.

**H2:** Collect as many seeds on the player's side.

**H3:** Reach as many moves as possible to select.

**H4:** Maximize the number of seeds in the player's kalaha.

**H5:** Clear the rightmost pit (i.e., that with number 1).

**H6:** Minimize the number of seeds in the opponent's kalaha.

The values of variables are natural numbers (including zero), except for $h_5 \in \{0, 1\}$.

Regarding H1 and H2, the left-most pit (number 6) is the best option to collect (hoard) seeds [81]. Heuristic H3 looks one move ahead to maximize the diversity of possible moves. Attacking heuristics H4 and H5 look one move forward, and they attempt to maximize the score (i.e., the number of stored seeds). Especially, empty pits on the right-hand side may lead to the possibility of repeated moves within the player's round [33,35], i.e., to achieve T4. For instance, even a simple scheme with 6–5–4–3–2–1 seeds in pits (from the left) may lead to 17 repeated moves (i.e., 17 seeds stored in the kalaha within a single round) while four seeds eventually remain in the player's pits. The chain of such moves starts with emptying the following pits (numbers): 1, 2, 1, 3, etc. Generally, it is desirable to reach a descending amount of seeds inside the pits starting from 6 in pit number 6, while some others on the right-hand side of the board may be empty. Or, a proper choice is to get an ascending number of seeds starting from 1 in the rightmost pit. In some cases, it is possible to clear all the pits at the player's side within a single round, which yields the end of the game. It was shown that the longest possible chain of such moves for the 6-hole board is 17 [56,75]. The initial scheme is 6–4–2–3–1–1 followed by the moves: 1, 3, 1, 2, etc. Note that heuristic H6 represents a defensive strategy looking two moves ahead.

The round-robin tournament in Awari and Oware showed that H6, H5, H1 are the strongest heuristics, while H3 is the weakest one [38]. Contrariwise, the values of $w_i$ indicate the strength of H4, H6, and H5. However, function (3) was set for more Mancala games simultaneously. These results indicate the advantage of the defensive strategy and asymmetric distribution of seeds inside the player's pits. Rovaris [26] played 200 games to benchmark the above-introduced heuristic approach against other algorithms. It was found that the evolved player wins 99.5% against the random player even for the tree-depth of 1, and it always wins for the depth of 8. It plays better than the greedy algorithm, yet the success rate does not depend on the tree-depth much. When playing against the standard mini–max of the depth of 1, the heuristic algorithm of the same depth wins 73.5%. If both the depths equal 4, the success rate is only 61%. Surprisingly, the advantage of the first player was not proved (the first player wins 73%, while as the second one 74%). The depth of 4 is comparable to the MCTS with approx. 4500 iterations. Finally, when playing against itself, the higher depth yields the higher win rate, yet the advantage of playing first is unclear.

It must be noted that the tree-search is still needed for this heuristic algorithm, which yields a longer computation time.

Unfortunately, function (3) does not include T4 that was shown as the strongest performing heuristics [33] since Awari analyzed in [38] does not incorporate T4 in its rules.

We also recall the works [40,64] where the importance of safe moves when perfectly solving was investigated. By adopting these ideas to Kalah, they agree with making moves that cannot be

attacked by the opponent without searching the game tree. It gives rise to ad-hoc hand-made heuristics provided by the current board positions for the players.

## 4. Heuristic-Deterministic Algorithm

Based on the game rules, their analysis, and the review provided in Section 3.2, the following rational suggestions giving rise to the eventual strategies and heuristics can be concluded:

1.  It is reasonable to apply the attacking tactic T4. In other words, a repeated move is preferable. However, its advantage decreases with the distance of the particular pit from the kalaha. Assume an empty pit near the kalaha. Moreover, let a left-hand side pit with a small number of seeds inside on the player's side exist, the sawing of which yields a repeated move. Then, the player has to be careful. Namely, if the player inserts a seed into the empty pit due to the sawing in the anti-clockwise direction, they can lose a chance to capture the opponent's seeds in the next turn.

2.  There is an advantage of the attacking heuristic H5 here as well. It means that it is suitable to clear the pit number 1. It is in contrast to H4 for the first sight. However, their combination can be paraphrased by the endeavor to use the sawing of the rightmost pit such that the last seed ends in the player's kalaha. Then, when sawing another pit, the rightmost one will have exactly one seed inside—therefore, a repeated move can be reached in the next turn.

3.  A defensive strategy is often neglected; however, heuristic H6 has been evaluated among the most powerful ones. The capturing represents the only possibility of how to increase the number of opponent's seeds, i.e., the number of potential seeds stored in their kalaha. Thus, the player has to check the existence of empty opponent's pits, into which the last seed can land within the next opponent move. This possible danger yields the necessity to analyze the opponent's move simultaneously.

A pseudo-code of the eventually proposed heuristic algorithm follows (Algorithm 1).

---
**Algorithm 1.** Heuristic

---
1 **function** Heuristic (*player*)
2      **if** LastSeedToKalaha (*player*) **return** True
3      **else if** LastSeedToTarget (*player*) **return** False
4      **else if** OpponentSeedToStash (*player*) **return** False
5      **else if** FirstNonEmptyPit (*player*) **return** False
6 **end function**

---

It can be seen that Algorithm 1 constitutes a hierarchy of a finite set of ordered functions based on the current game state. Function HEURISTIC returns True only if a repeated move within the current turn is possible. If the particular function on a certain level returns True, the corresponding game move is made. Pseudo-codes and concise descriptions of these functions, which perform the moves, follow.

---
1 **function** LastSeedToKalaha (*player*)
2      *help* ← False
3      *pitNum* ← 1
4      **for** all *player*'s pits (*pitNums*) **do**
5          **if** the last seed of the pit with number *pitNum* lands in kalaha **then**
6              Sawing (player, pitNum)
7              *help* ← True
8              *pitNum* ← 1
9          **else**
10              *pitNum* ← *pitNum* + 1
11          **end if**
12      **end for**
13      **return** *help*
14 **end function**

---

The function checks whether it is possible for the player to place the last sawed seed from a particular pit (determined by its number *pitNum*) into the kalaha. If it is possible, Sawing function ensures the sawing itself for the pit.

---

1 **function** LastSeedToTarget (*player*)
2   $i \leftarrow 1$
3   $pitNum \leftarrow 1$
4   $j \leftarrow 1$
5   $value \leftarrow 0$
6   **for** all *player*'s pits with numbers *i* **do**
7     **if** (pit with number *i* is empty) AND (the number of seeds in the opposite pit > *value*)
8     **then**
9       **for** all *j* > *i* **do**
10         **if** the last seed from *j* lands into *i* **then**
11           $pitNum \leftarrow j$
12           $value \leftarrow$ number of seeds in the pit opposite to *i*
13           **break**
14         **end if**
15       **end for**
16     **end if**
17     **else if** (pit *i* has 13 seeds) AND (the number of seeds in the opposite pit > (*value* − 1))
18     **then**
19       $pitNum \leftarrow i$
20       $value \leftarrow$ number of seeds in the pit opposite to *i* + 1
21     **end if**
22     **else if** (an empty pit with number *j* > *i* exists such that the last seed from *i* lands to *j*)
23         AND (the number of seeds in the pit opposite to *j* > (*value* − 1)) **then**
24       $pitNum \leftarrow i$
25       $value \leftarrow$ number of seeds in the pit opposite to *j* + 1
26     **end if**
27   **end for**
28   **if** *value* > 0 **then**
29     Sawing (player, pitNum)
30     **return** True
31   **else**
32     **return** False
33   **end if**
34 **end function**

---

The function attempts to find a pit, the sowing of which ensures that some opponent's seeds can be captured. Three different scenarios are tested. First, an empty pit with a non-empty number of seeds in the opposite hole is searched for. Second, a possible roundabout move finishing in the starting pit is tested. Third, the existence of another type of roundabout is investigated where the last seed of the sawing ends in a pit left from the starting one. The variable value indicates how many opponent's seeds are captured. If no capture is possible, the function returns False.

Function OPPONENTSEEDTOSTASH works analogously to LASTSEEDTOTARGET, yet from the opponent's perspective. It searches for a player's pit that is the most threatened by the capturing. Then, seeds from this pit are sawed. However, such a defensive move may yield a loss of the player's opportunity to capture the opponent's seeds in the next turn.

```
1 function FirstNonEmptyPit (player)
2      pitNum ← 1
3      for all player's pits (pitNum) do
4          if the pit is non-empty then
5              Sawing (player, pitNum)
6              return True
7          end if
8      end for
9      return False
10 end function
```

This function of the lowest priority inside the heuristic algorithm simply searches for a non-empty pit with the lowest possible pit number (i.e., starting from the player's right-hand side of the board). If such a pit exists, it is sawed; otherwise, the game ends.

## 5. Modified Game-Tree Search Algorithm

The proposed heuristic algorithm is benchmarked with the standard mini–max game-tree search algorithm [26,41] with alpha-beta pruning and its modification that brings about a dash of heuristics. Let us recall the standard mini–max search first. Let us recall the standard mini–max search first (Algorithm 2).

In the algorithm, constant MaxDepth expressing the maximum depth (level) of the tree search is a priori given. The variable board means the current state of the board, i.e., the root node of the tree. All the available moves are considered to be made from this node. The player searches for the maximum value of the cost function Evaluate; whereas, the opponent calculates its minimum.

According to Donkers [73], the evaluation (cost) function returns a scalar value that measures the strength of that position of the player. The function returns the so-called heuristic value whenever the current board position of the game is not terminal. There exist three types of cost functions and their heuristic values here. They can express one of the following terms:

1. A predictor for the true game-theoretic value of a position. The higher the heuristic value for a position is, the more certain is that the position yields a game-theoretic win.
2. The probability of winning the game at that position. Authors of machine-learning techniques use this interpretation.
3. A measure of the profitability of the position. The concept is used in many knowledge-based, hand-made heuristic evaluation functions.

Moreover, it is stressed in [73] that the simultaneous evaluation of both the player's and the opponent's moves is desirable. Therefore, the cost function has to assess the board state at the end of both turns. Oon and Lim [69] state that Kalah tends to confer a significant advantage to the first player. Therefore, the use of an evaluation function considering only moves of one player may be unable to play well for both the player and the opponent.

Thus, we have constructed the following cost function, the meaning of which agrees with the last item:

$$f(p) = 10\left[\left(N_{p,end} - N_{p,start}\right) - \left(N_{o,end} - N_{o,start}\right)\right] + N_{p,rep} + 5N_{p,1}. \tag{4}$$

If Formula (4) is taken from the player's ($p$) perspectives, the opponent's data are denoted by the subscript $o$. Values of $N_{\cdot,start}$, $N_{\cdot,end}$ mean the initial and the final number of seeds stored in the particular kalaha within the given MaxDepth, respectively. The eventual number of stored seeds at the end of the game decides about the winner. Therefore, these values are of the highest importance. The value of $N_{p,rep}$ expresses a less-significant data about the number of repeated moves. Such moves bring about a certain advantage for the player. Finally, the Boolean variable $N_{p,1}$ returns 1 if the pit number 1 is empty at the end of the selected tree-searching window; otherwise, it is set to 0. The scenario $N_{p,1} = 1$

means a high advantage for the player to place a seed inside the pit 1 in the next turn. This sawing yields either a possibility to capture some opponent's seed within the turn or to reach a repeating move during the after next turn. The opponent uses Equation (4) analogously. Note that the scaling factors have been selected intuitively according to their importance. There is, naturally, an open research task of their optimization, which goes beyond the topic of this paper.

---

**Algorithm 2.** Standard mini–max

---

**1 function** StandardMinimax (*player, board, alpha, beta, depth*)

2      **if** (*depth* = MaxDepth) or (*board* means end of the game) **then**

3          **return** Evaluate (*player*, *board*)

4      **end if**

5      **if** *player* = Player **then**

6          *bestValue* ← −∞

7          **for** all Player's pits with numbers *pitNo* **do**

8              *board* ← Sawing (*Player, pitNo*)

9              **if** a repeated move is possible **then**

10                 *value* ← StandardMinimax (Player, *board, alpha, beta, depth*)

11              **else**

12                 *value* ← StandardMinimax (Opponent, *board, alpha, beta, depth* + 1)

13              **end if**

14              *bestValue* ← Max (*value, bestValue*)

15              *alpha* ← Max (*alpha, bestValue*)

16              **if** *beta* ≤ *alpha* **then**

17                 **break**

18              **end if**

19          **end for**

20      **else**

21          *bestValue* ← ∞

22          **for** all Opponent's pits with number *pitNo* **do**

23              *board* ← Sawing (*Opponent, pitNo*)

24              **if** a repeated move is possible **then**

25                 *value* ← StandardMinimax (Opponent, *board, alpha, beta, depth*)

26              **else**

27                 *value* ← StandardMinimax (Player, *board, alpha, beta, depth* + 1)

28              **end if**

29              *bestValue* ← Min (*value, bestValue*)

30              *beta* ← Min (*beta, bestValue*)

31              **if** *beta* ≥ *alpha* **then**

32                 **break**

33              **end if**

34          **end for**

35      **return** *bestValue*

36 **end function**

---

The presented basic mini–max algorithm is a sufficient benchmark representative to test our heuristic algorithm (Algorithm 1). Indeed, as mentioned in Section 2, Ramanujan [62] concluded that the use of some advanced game-tree search algorithms brings no evident enhancement compared to the basic algorithm for some Mancala games. Nevertheless, we have decided to combine selected heuristic strategies used in Algorithm 1 with Algorithm 2, giving rise to a modified (mini–max) game-tree search Algorithm 3—that serves as another benchmark. Its concise pseudo-code follows.

---

**Algorithm 3.** Modified mini–max

| | |
|---|---|
| 1 | **function** MODIFIEDMINIMAX (*player, board, alpha, beta, depth*) |
| 2 |  *help* ← False |
| 3 |  **if** (*depth* < MaxDepth) AND (*board* includes any non-empty pit on the *player*'s side) **then** |
| 4 |   **if** LASTSEEDTOKALAHA (*player*) **then** |
| 5 |    Update *board* |
| 6 |    MODIFIEDMINIMAX (*player, board, alpha, beta, depth*) |
| 7 |    *help* ← True |
| 8 |   **end if** |
| 9 |   **else if** LASTSEEDTOTARGETSIMPLE (*player*) **then** |
| 10 |    Update *board* |
| 11 |    *help* ← True |
| 12 |   **end if** |
| 13 |   **if** not(*help*) **then** |
| 14 |    STANDARDMINIMAX (*player, board, alpha, beta, depth*) |
| 15 |   **end if** |
| 16 |  **end if** |
| 17 | **end function** |

---

where function LASTSEEDTOTARGETSIMPLE is a simplified function LASTSEEDTOTARGET that ignores the pit number 1 and empty opposite opponent's pits when searching for a target. Moreover, it does not consider a possible roundabout when searching for a pit, the last sawed seed from which lands into the target.

---

| | |
|---|---|
| 1 | **function** LASTSEEDTOTARGETSIMPLE (*player, board, depth*) |
| 2 |  *i* ← 1 |
| 3 |  *value* ← 0 |
| 4 |  *pitNum* ← 1 |
| 5 |  **for** all *player*'s pits with numbers *i* **do** |
| 6 |   **if** (pit with number *i* is empty) **then** |
| 7 |    **if** (*i* = 0) OR (the opposite pit is empty) **then** |
| 8 |     *i* ← *i* +1 |
| 9 |    **else for** all *j* > *i* **do** |
| 10 |     **if** the last seed from *j* lands into *i* **then** |
| 11 |      *pitNum* ← *j* |
| 12 |      *value* ← number of seeds in the pit opposite to *i* |
| 13 |      **break** |
| 14 |     **end if** |
| 15 |    **end for** |
| 16 |   **end if** |
| 17 |   **end if** |
| 18 |  **end for** |
| 19 |  **if** *value* > 0 **then** |
| 20 |   SAWING (*player, pitNum*) |
| 21 |   **return** True |
| 22 |  **else** |
| 23 |   **return** False |
| 24 |  **end if** |
| 25 | **end function** |

---

The use of the two heuristic strategies in Algorithm 3 should bring the advantage of faster and partially knowledge-based, not the full brute-force, computation of a suitable player's move.

Note that STANDARDMINIMAX in line 14 of Algorithm 3 is modified such that it calls MODIFIEDMINIMAX in lines 10, 12, 25, 27 of Algorithm 2 instead of STANDARDMINIMAX.

To sum up, function MODIFIEDMINIMAX attempts to make safe moves, i.e., those not causing any loss for the player. Moreover, a searching effort is considerably reduced, which is close to the idea of perfect solving [40].

It is worth noting that even if the game-tree search algorithm finds the best move based on the heuristic value of the cost function (for the given MaxDepth), the value can be far from the game-theoretic one. For instance, Irving et al. [29] already proved that if the player stores six seeds in their kalaha during the first six turns, they definitely lose the game. However, if they capture only two seeds, they have over 13,000 further winning move possibilities, while only approx. 8000 lead to the loss. Generally, these paradoxes give rise to the notion of game-tree pathology [1]. Nau et al. [82] showed that Kalah is the first real game to exhibit pathology throughout the game consistently. Note that an error-minimizing mini–max algorithm was proposed by Zuckerman et al. [1] to overcome the effect of pathology.

## 6. Software Implementation

Let us now concisely provide the reader with the GUI that has been developed to verify and benchmark the proposed heuristic algorithm, and that allows users to play against the algorithms. Since the aims of this paper are to provide a survey on Kalah and to propose novel heuristic strategies, it is worth noting that only a necessary effort has been made to develop the GUI, which does not meet a graphical attractiveness. The programmed application is not intended to be publicly or commercially released.

There exist several free or commercial applications and other software solutions that allow users to play the Kalah game. We do let mention just a few. Mancala Ultimate [83] by MobileFusion Apps Ltd. enables the selection of one of three modes—man–man played on a single machine or via the internet, and a man-computer mode. However, some users complain that only one mode is enabled, and it is too easy to beat the computer since it repeats the same predictable moves. The application by AppOn Innovate [84] enables us to play the Mancala game in the on-line or the off-line modes against a real or an artificial player. Yet, it is unstable sometimes, and the human player has to wait for a move by the machine for tens of minutes. Kalah Mancala by Purple Squirrel Productions [85] enables the user to play against 12 characters (with a different behavior); however, no application updates have been made until 2017. For this research, an interesting application can be found at www.mathplayground.com [86] that implements a smart and sophisticated algorithm. However, it enables us to play Kalah on-line only in the man-machine mode. Another issue is that most of the existing implementations do not enable us to determine who or what starts to play. This option is, however, very important [29,32,36–38].

C++ is a well-known, universal, and object-oriented programming language. Software Development Kits (SDKs) and frameworks—i.e., sets of library modules and other programming tools—Extend the language towards easy-to-develop software applications. A combination of C++ and SDK called Qt [87] has been used for our GUI application. Qt provides a software developer with the necessary tools for application design and deployment and is characterized by the usage of signals and slots. A signal is sent by an object (e.g., a widget) when some event occurs. A slot means a function that returns an answer to the signal.

When the user opens the application, a GUI window is released, through which the first player be chosen, see Figure 6a. It can be further set what algorithm is applied on the computer side: Either the heuristic strategy given by Algorithm 1 or the modified game-tree search mini–max (according to Algorithm 3) can be selected. The latter option enables us to set the desired value of MaxDepth, see Figure 6b. Note that Algorithm 2 (i.e., the standard mini–max) is not implemented in GUI since it serves solely to benchmark testing and can be easily obtained by altering some parts of Algorithm 3.

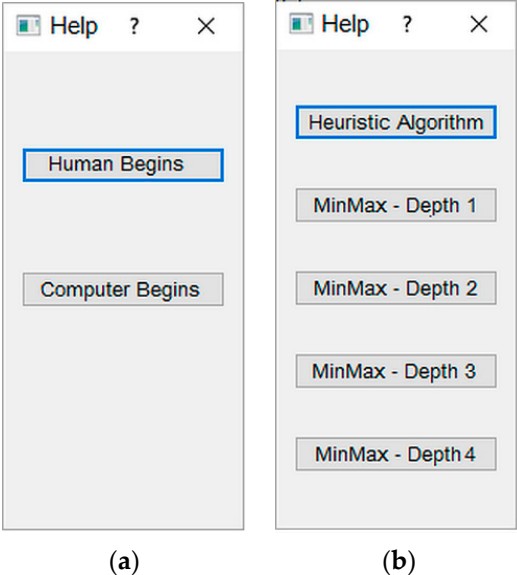

(**a**)    (**b**)

**Figure 6.** The introduction window (**a**) and the algorithm-selection window(**b**) of the GUI.

Once the algorithm is selected, the main window is opened, see Figure 7a. The human player (i.e., the player) always holds the south side of the board. The pit that should be sawed is simply selected by double-clicking. When the game is over, a window displaying results is released, see Figure 7b. The user can further choose whether to play again or to quit the application.

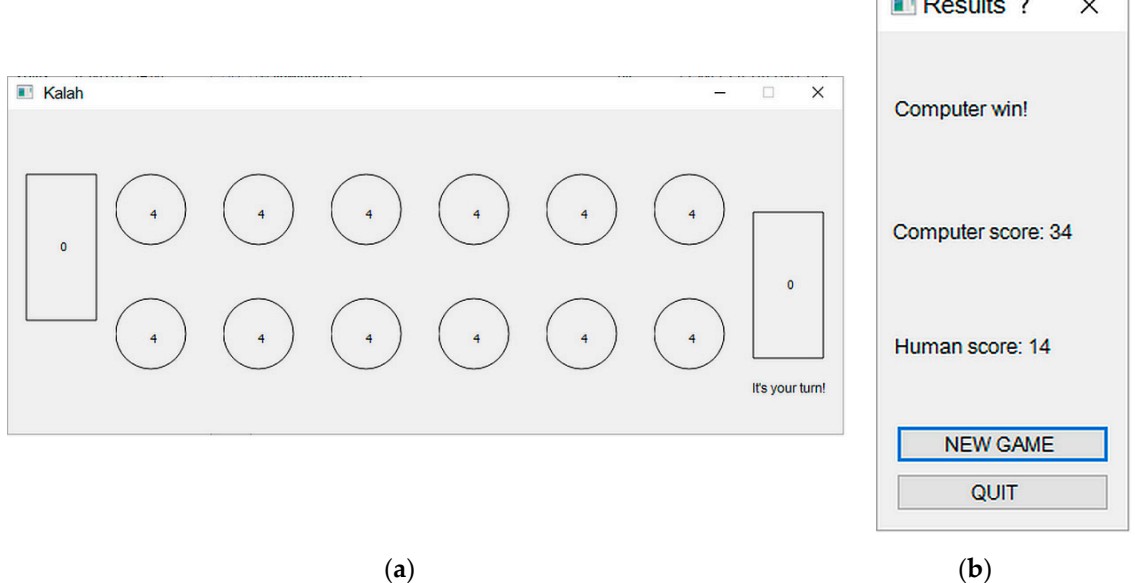

(**a**)    (**b**)

**Figure 7.** The main window (**a**) of the GUI and the window with game results (**b**).

## 7. Experimental Results and Evaluation

Two sets of benchmark tests follow. A tournament where a mid-experienced amateur human player competes with the three algorithms is introduced first. Then, a round-robin tournament of all the algorithms is presented. It is worth noting that heuristic Algorithm 1 is the center of attention here. The particular result quality is measured primarily by the number of seeds stored in the kalaha and also by the computational time, as an auxiliary measure. The GUI runs on Intel© Core™ i3-4000M CPU @ 2.39 GHz with 4 GB RAM.

Every pair of rivals played 20 games with MaxDepth = 1, 2, 4, 6 in the former set of tests. The computer played half of the games as first and vice versa. In the latter benchmark, only two games were computed for every pair of competing algorithms. Again, every algorithm starts as the first player for half of the games.

### 7.1. Computer vs. Human Player

Table 1 displays the results of the tournament where the computer (i.e., the proposed and introduced algorithms) as opponent played against the human player (i.e., player). The values inside the table express the number of stored seeds for each of the players. The evaluation of the results can be found in Section 7.3.

**Table 1.** Scores of the human player and the computer.

| Algorithm No. | 1st Player | Score for | Number of the Game | | | | | | | | | |
|---|---|---|---|---|---|---|---|---|---|---|---|---|
| | | | 1 | 2 | 3 | 4 | 5 | 6 | 7 | 8 | 9 | 10 |
| 1 | Human | Human | 38 | 27 | 36 | 40 | 33 | 41 | 24 | 26 | 34 | 29 |
| | | Computer | 10 | 21 | 12 | 8 | 15 | 7 | 24 | 22 | 14 | 19 |
| | Computer | Human | 19 | 13 | 19 | 19 | 19 | 15 | 20 | 16 | 23 | 16 |
| | | Computer | 29 | 35 | 29 | 29 | 29 | 33 | 28 | 32 | 25 | 32 |
| 2 (MaxDepth 1) | Human | Human | 33 | 35 | 31 | 21 | 36 | 27 | 39 | 28 | 34 | 23 |
| | | Computer | 15 | 13 | 17 | 27 | 12 | 21 | 9 | 20 | 14 | 25 |
| | Computer | Human | 18 | 16 | 7 | 16 | 12 | 15 | 10 | 25 | 22 | 26 |
| | | Computer | 30 | 32 | 41 | 32 | 36 | 33 | 38 | 23 | 26 | 22 |
| 2 (MaxDepth 2) | Human | Human | 27 | 20 | 24 | 23 | 11 | 24 | 26 | 29 | 22 | 30 |
| | | Computer | 21 | 28 | 24 | 25 | 37 | 24 | 22 | 19 | 26 | 18 |
| | Computer | Human | 22 | 10 | 17 | 12 | 17 | 15 | 23 | 21 | 16 | 6 |
| | | Computer | 26 | 38 | 31 | 36 | 31 | 33 | 25 | 27 | 32 | 42 |
| 2 (MaxDepth 4) | Human | Human | 20 | 29 | 19 | 11 | 17 | 24 | 21 | 26 | 16 | 25 |
| | | Computer | 28 | 19 | 29 | 37 | 31 | 24 | 27 | 22 | 32 | 23 |
| | Computer | Human | 7 | 4 | 5 | 6 | 6 | 5 | 8 | 7 | 6 | 11 |
| | | Computer | 41 | 44 | 43 | 42 | 42 | 43 | 40 | 41 | 42 | 37 |
| 2 (MaxDepth 6) | Human | Human | 13 | 12 | 18 | 11 | 26 | 24 | 17 | 12 | 12 | 28 |
| | | Computer | 35 | 36 | 20 | 37 | 22 | 24 | 31 | 34 | 36 | 20 |
| | Computer | Human | 6 | 6 | 4 | 4 | 7 | 5 | 9 | 5 | 6 | 6 |
| | | Computer | 42 | 42 | 44 | 44 | 41 | 43 | 39 | 43 | 42 | 42 |
| 3 (MaxDepth 1) | Human | Human | 35 | 34 | 22 | 32 | 20 | 29 | 35 | 22 | 16 | 35 |
| | | Computer | 13 | 14 | 26 | 16 | 28 | 19 | 13 | 26 | 32 | 13 |
| | Computer | Human | 18 | 16 | 7 | 16 | 12 | 15 | 10 | 25 | 22 | 26 |
| | | Computer | 30 | 32 | 41 | 32 | 36 | 33 | 38 | 23 | 26 | 22 |
| 3 (MaxDepth 2) | Human | Human | 23 | 23 | 28 | 25 | 25 | 25 | 10 | 29 | 24 | 32 |
| | | Computer | 25 | 25 | 20 | 23 | 23 | 23 | 38 | 19 | 24 | 16 |
| | Computer | Human | 17 | 18 | 12 | 13 | 23 | 19 | 13 | 15 | 7 | 13 |
| | | Computer | 31 | 30 | 36 | 35 | 25 | 29 | 35 | 33 | 41 | 34 |
| 3 (MaxDepth 4) | Human | Human | 22 | 16 | 31 | 19 | 20 | 24 | 22 | 25 | 18 | 26 |
| | | Computer | 26 | 32 | 17 | 29 | 28 | 24 | 26 | 23 | 30 | 22 |
| | Computer | Human | 5 | 9 | 6 | 5 | 5 | 7 | 7 | 5 | 6 | 8 |
| | | Computer | 40 | 39 | 41 | 35 | 33 | 41 | 41 | 43 | 41 | 40 |
| 3 (MaxDepth 6) | Human | Human | 18 | 19 | 16 | 24 | 20 | 12 | 18 | 26 | 16 | 20 |
| | | Computer | 30 | 29 | 32 | 24 | 28 | 36 | 30 | 22 | 32 | 28 |
| | Computer | Human | 4 | 4 | 7 | 6 | 6 | 5 | 5 | 6 | 8 | 7 |
| | | Computer | 44 | 44 | 41 | 42 | 42 | 43 | 43 | 42 | 40 | 41 |

### 7.2. Algorithm vs. Algorithm

The results of the round-robin tournament of the algorithms are given to the reader in Table 2. The first column indicates what the algorithm plays first. The value of MaxDepth is given in brackets. Again, the results are evaluated in Section 7.3.

**Table 2.** Results of the round-robin tournament of the algorithms.

| Algorithm Playing First | 1 | 2 (1) | 2 (2) | 2 (4) | 2 (6) | 3 (1) | 3 (2) | 3 (4) | 3 (6) |
|---|---|---|---|---|---|---|---|---|---|
| 1 | 32:16 | 33:15 | 31:17 | 18:30 | 16:32 | 34:14 | 33:15 | 20:38 | 17:31 |
| 2 (1) | 28:20 | 22:26 | 26:24 | 41:7 | 39:9 | 29:19 | 25:23 | 36:12 | 38:10 |
| 2 (2) | 31:17 | 32:16 | 24:24 | 41:7 | 40:8 | 33:15 | 29:19 | 44:4 | 42:6 |
| 2 (4) | 45:3 | 37:11 | 41:7 | 41:7 | 41:7 | 44:4 | 43:5 | 44:4 | 44:4 |
| 2 (6) | 45:3 | 39:9 | 41:7 | 41:7 | 41:7 | 44:4 | 43:5 | 44:4 | 44:4 |
| 3 (1) | 26:22 | 28:20 | 27:21 | 38:10 | 38:10 | 24:24 | 29:19 | 42:6 | 41:7 |
| 3 (2) | 30:18 | 32:16 | 38:10 | 40:8 | 40:8 | 33:15 | 28:20 | 42:6 | 42:6 |
| 3 (4) | 44:4 | 36:12 | 40:8 | 40:8 | 40:8 | 39:9 | 42:6 | 42:6 | 42:6 |
| 3 (6) | 44:4 | 38:8 | 40:8 | 40:8 | 40:8 | 40:8 | 40:8 | 42:6 | 42:6 |

### 7.3. Results Evaluation

Let us evaluate Table 1 first. The concern is centered on the win rate, the average number captured seeds, the computing time, and the advantage of the opening move. Figures 8–10 clearly display these statistic data where the left-hand side ones belong to games that are opened by the human player. When computing the win rate for the computer, a victory has a weight of 2, a draw is weighted by 1, and a loss by 0. Note that the charts for Algorithms 2 and 3 coincide in Figure 8. The observed computing time (per a move) mean values in the logarithmic scale are given in Figure 10. In all the figures, the constant line indicates the particular value for Algorithm 1.

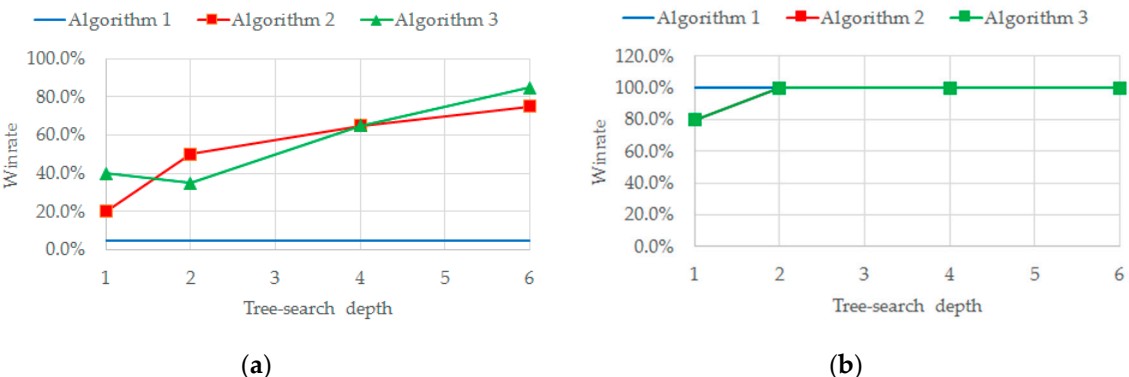

(**a**)        (**b**)

**Figure 8.** Winrates for the algorithms when the opening move is made by the human player (**a**) and the computer (**b**).

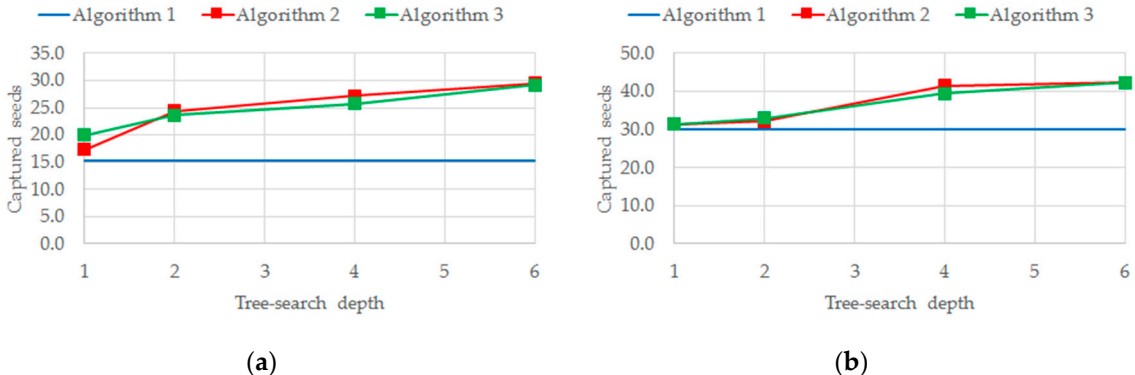

(**a**)        (**b**)

**Figure 9.** Numbers of seeds captured by the algorithms when the opening move is made by the human player (**a**) and the computer (**b**).

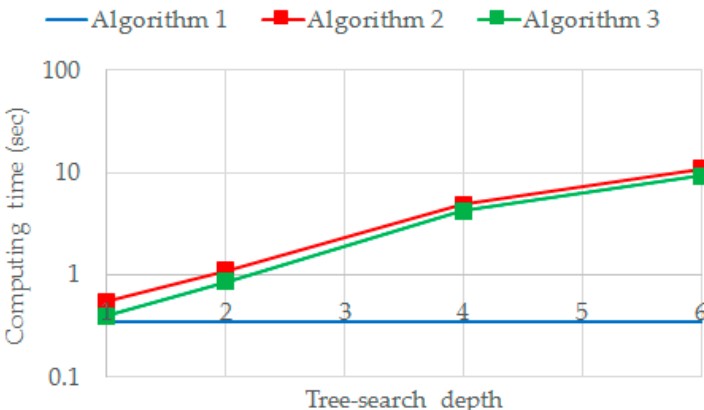

**Figure 10.** Average computing time per move decision made by the algorithms.

Figure 8a,b display "the quality of playing". It can be observed that the order of the opening move has a decisive impact on the win for Algorithm 1 and Algorithms 2 and 3 with MaxDepth of 1. If the human player starts to play as first for MaxDepth = 2, a tough contest can be followed. For a deeper game-tree search, both the mini–max algorithms play equal or better regardless of what or who plays first. Concerning the heuristic algorithm (Algorithm 1), it managed to win all the games when playing first. Contrariwise, it cannot beat the human opponent if it started second, except for one tie game. Algorithm 1 and Algorithm 2 with MaxDepth = 1 thus give symmetrical results when playing first and second against a human player.

Figure 9a,b display "the quantity of playing". The average number of collected seeds by the human player when they open the game was 32.8, whereas the computer captured 30.1 when it opens the game. Although there is a little shift towards the human player, one can conclude that Algorithm 1 plays almost as good as a mid-experienced amateur player. For Algorithms 2 and 3 with MaxDepth of 1, it was 30.7:31.3 and 28.0:31.3, respectively. Hence, these algorithms have similar success to Algorithm 1 if the computer plays first.

As can be observed from Figure 10, the processing time required for the move decision highly increases with the tree depth. Whereas it ranges from 0.2 to 0.5 s for Algorithm 1, the ranges for Algorithm 2 are [0.3, 0.8], [0.8, 1.4], [3.3, 6.5], [5.9, 15.7] seconds for MaxDepth = 1, 2, 4, 6, respectively, and these are [0.2, 0.6], [0.6, 1.1], [2.7, 5.7], [4.3, 14.1] seconds for Algorithm 3. Hence, due to heuristics used in Algorithm 3, this time is shorted a bit on average compared to Algorithm 2. The authors mean that higher values of MaxDepth are not suitable for real-life human-machine interaction. For instance, if Algorithm 2 has to check and evaluate six game-tree levels for both the players, it requires to compute up to $6\left(7 + \sum_{i=2}^{11} 6^i\right) = 2{,}612{,}138{,}802$ nodes (without the pruning and repeated moves).

To judge the obtained results for the human vs. the computer more rigorously, we do let perform statistical induction methods for paired data at the given significance level 0.05. The difference between the computer's score and the human's score is considered. That is, a positive value means the superiority of the computer. As first, it has to be verified whether the data sets are normally distributed by using the Shapiro–Wilk test [88]. Since it can be deduced that some of the data are not normally distributed, nonparametric methods are to be used. Namely, the confidence intervals (CI) of medians of score differences and the Wilcoxon two-tailed (signed rank) test for paired samples instead of the t-test are performed. The obtained results are summarized in Table 3.

From Table 3, it is apparent that, e.g., if the human player competes against Algorithm 1 and makes the opening move, they win by at least 19 for half of the games. In addition, it can be estimated (for the selected confidence level 0.95) that the median of score differences varies from 10 to 29 in favor of the human. A statistically significant result in the score differences can also be deduced from the Wilcoxon test (*p*-value = 0.009).

**Table 3.** Statistical tests on selected data from Table 1.

| Algorithm | 1st Player | Median (95% CI) | Wilcoxon Signed Rank Test (*p*-Value) | Better Player |
|---|---|---|---|---|
| 1 | Human | −19 (−29, −10) | 0.009 | Human |
| | Computer | 13 (9, 16) | 0.005 | Computer |
| 2 (1) | Human | −14 (−22, −4) | 0.017 | Human |
| | Computer | 15 (5, 25) | 0.016 | Computer |
| 2 (2) | Human | −1 (−9, 14) | 0.889 | Draw |
| | Computer | 16 (8, 25) | 0.006 | Computer |
| 3 (1) | Human | −8 (−20, 5) | 0.113 | Draw |
| | Computer | 15 (5, 25) | 0.016 | Computer |
| 3 (2) | Human | −2 (−9, 13) | 0.399 | Draw |
| | Computer | 18 (11, 24) | 0.006 | Computer |

To sum up, the first player shows a statistically significantly better result in cases of Algorithm 1 and Algorithm 2(1). In cases of Algorithm 2(2) and Algorithm 3, the computer algorithms report significantly better results if they make the opening move. Contrariwise, if the human player opens the game, there is no statistically significant difference between win successes of the human player and the computer. Note that clear cases are not included in the tests.

Regarding Table 3, numbers of stored seeds for Algorithm 1 and Algorithm 2 (which open the game) against other algorithms, and Algorithm 1 with Algorithm 3 against all the others are displayed in Figures 11 and 12, respectively.

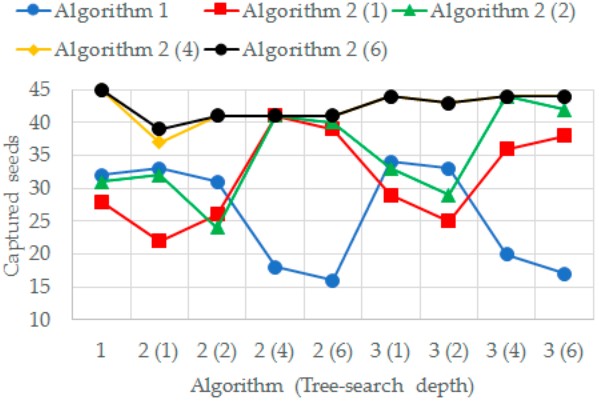

**Figure 11.** Numbers of seeds captured by Algorithms 1 and 2 when they open the game in the round-robin tournament.

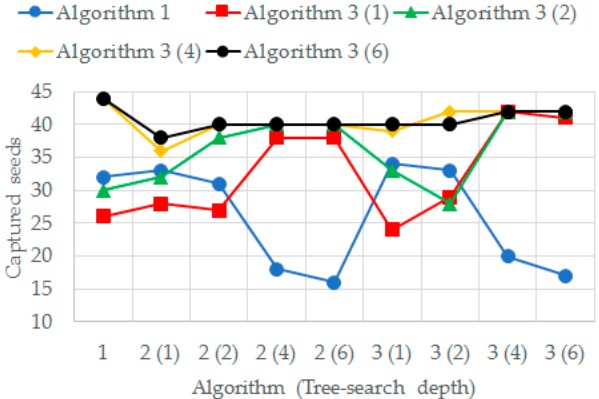

**Figure 12.** Numbers of seeds captured by Algorithm 1 and Algorithm 3 when they open the game in the round-robin tournament.

It can be deduced from Figure 11 that Algorithm 1 can beat all other ones when playing as first, for the MaxDepth of 1 and 2. However, it always loses when playing as the second. Again, it can be concluded the success of Algorithm 1 is about those of Algorithms 2 and 3 with the depth of 2, yet it is slightly better than the MaxDepth = 1. It means that it can serve as a faster and comparable alternative to a simple (but slavish) game-tree search. By comparing results for Algorithm 2 and Algorithm 3, one can observe that Algorithm 3 favors the first player a bit more. It might be caused by the heuristic strategies implemented in the algorithm. Another peculiar issue is that even if a tree-search algorithm plays against another one with a higher depth of search, it always wins when playing as first. Moreover, the stronger the second player is, the more substantial the victory of the first player is (except for the case of MaxDepth = 2 where local minima appear surprisingly). Last but not least, the higher the depth used by one of the algorithms is, the lower the difference between results for close depths is. In other words, the increase of MaxDepth leads to a convergence of the game result when competing Algorithms 2 and 3.

We do let provide the reader with suitable statistical tests again. To verify the difference of quality for Algorithms 1, 2(1), 2(2), 3(1), and 3(2), a comparison of scores reached by particular algorithms when playing against all the other (eight) ones is made, see Table 2 and Figure 13. The Shapiro–Wilk test verifies the normal distribution, and Bartlett's test was applied to decide the homoscedasticity [88]. The ANOVA test shows that there exist statistically significant differences at the confidence level $\alpha = 0.05$ ($p$-value = 0.006). According to the multiple comparisons via the Tukey HSD method, it was proved that Algorithms 2(2) and 3(2) get statistically better scores compared to other algorithms.

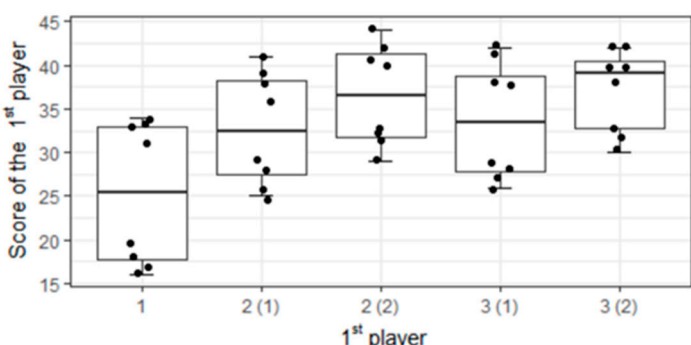

**Figure 13.** Box plots with jitter points for the 1st player in Table 2.

Recall that the game-theoretic value for the very standard version of Kalah yields a win by 10 for the first player [29] (i.e., the winner eventually stores 29 seeds). Notice that this expectation was met for the same cases in Table 3 for Algorithms 2 and 3 with MaxDepth of 1 or 2.

## 8. Conclusions

The primary aim of this paper has been to provide the reader with an overview of Kalah (Mancala games) history, solution of the game, and the research made. The knowledge of these board games and their state of the art can contribute to the world heritage since the origin of the family of Mancala games dates thousands of years back.

This survey indicated a lack of knowledge-based and artificial-intelligence-based strategies. Brute-force methods have predominated so far due to the high solution size of the game. Therefore, we have proposed a simple heuristic algorithm, the strategies of which are based on a thorough analysis of the Kalah game rules. In contrast to brute-force methods, the algorithm brings about an ordered set of ad-hoc strategies that are close to thinking of a real human player. Besides, it does not require a game-tree search and saves the computation time. The aim has been to get closer to rational thinking when playing.

In addition to that, the habitual mini–max game-tree search algorithm with alpha–beta pruning has been introduced. We have also proposed its modification implementing some save heuristic moves that save computational time in many cases. Via a programmed simple GUI application, it has been shown that the proposed heuristic algorithm gives results comparable to a mid-experienced human player. Moreover, the proposed heuristic algorithm offers results similar to both tree-search solutions. Experiments further proved that the order in which players start has had a decisive impact on the win rate. The proposed algorithm could beat the tree-search ones up to the search-depth of 2. The heuristic algorithm should implement a kind of self-learning strategy during the game to reach better performance.

In the future, research on a more sophisticated heuristics based on the opponent-based model and self-learning can represent an open task. An analysis of human thinking about moves in the Kalah game may help the computer to find their matter of weakness, and further contribute to cultural heritage.

**Author Contributions:** Conceptualization, L.P. and R.M.; methodology, L.P.; software, J.A.; validation, L.P. and J.A.; formal analysis, L.P., M.L.; investigation, L.P.; resources, L.P. and J.A.; data curation, L.P.; writing—original draft preparation, L.P. and R.M.; writing—review and editing, L.P. and R.M.; visualization, L.P.; supervision, L.P.; project administration, L.P. All authors have read and agreed to the published version of the manuscript.

**Funding:** The authors would like to express their gratitude to the European Regional Development Fund and to the Ministry of Education, Youth and Sports that financially supported this work under the National Sustainability Programme project No. LO1303 (MSMT-7778/2014) and to the internal grant agency of VSB Technical University of Ostrava, Faculty of Electrical Engineering and Computer Science, Czech Republic, under the project no. SP2020/46.

**Conflicts of Interest:** The authors declare no conflict of interest.

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
