# Peer review of "Review of Kalah Game Research and the Proposition of a Novel Heuristic–Deterministic Algorithm Compared to Tree-Search Solutions and Human Decision-Making"

_informatics, doi:10.3390/informatics7030034_

Round 1
Reviewer 1 Report
The new title is really long, reflecting the attempt of the
paper to be both survey and research article. The presentation
is improved. The second paragraph in Section 8 Conclusions
need to be adjusted (sentence structures).
I recommend acceptance and I think this will be a quite
unconventional paper with both its survey content and its
algorithmic contributions.
Author Response
Thank you very much again for your time and effort in coordinating the review process of informatics-913648 manuscript and for giving us a chance to revise and resubmit the manuscript. We still believe that the given topic and results are suitable for inclusion in your journal and attractive enough to the reader.
Comments by both the reviewers have been addressed in the revised version. In the revised manuscript, the modifications according to comments from Reviewer 1 are all highlighted in blue.
Reply to Reviewer 1
Many thanks to this reviewer for their comments. We have attempted to revise the paper accordingly.
1) Comment: The new title is really long, reflecting the attempt of the paper to be both survey and research article.
Response: Thank you for identifying the reason for such a long title. We would like to cover both the main parts of the manuscript content; therefore, the title seems to be wordy. However, have eventually slightly altered the title to Review of Kalah Game Research and the Proposition of a Novel Heuristic-Deterministic Algorithm Compared to Three-Search Solutions and Human Decision-Making. Hence, the current title has one word less than the previous one.
2) Comment: The presentation is improved.
Response: Your positive comment is much appreciated.
3) Comment: The second paragraph in Section 8 Conclusions need to be adjusted (sentence structures).
Response: Thank you for pointing out this issue. We have reformulated the paragraph accordingly.
Reviewer 2 Report
I liked the refocus of the paper to be a survey paper on Kalah rather than an algorithmic focus. However, the abstract is not focused and the last couple of sentences does not make a sense and does not connect well to the first part logically (e.g. the last two sentences does not make sense). I suggest rewriting the abstract.
My comment on the statistical results is still standing, even with the new changes. I do not think that 20 games will allow for a statistical significance difference between the algorithms. Anyway, regardless of my own intuition, statistical test should be applied. The win rate difference when only 20 games were played is not enough, especially when apparent differences are so small.
It seems that the authors took the time and effort to address all the comments in a satisfactory way. The only other question to consider is whether the (refocused) topic of the paper is adequate to the journal’s aim and scope, but that is the editor’s job to decide.
Author Response
Thank you very much again for your time and effort in coordinating the review process of informatics-913648 manuscript and for giving us a chance to revise and resubmit the manuscript. We still believe that the given topic and results are suitable for inclusion in your journal and attractive enough to the reader.
In the revised manuscript, the modifications according to comments from Reviewer 2 are highlighted in red.
Reply to Reviewer 2
Thank you very much for your pertinent comments and also for recognizing our effort to revise the original manuscript thoroughly.
1) Comment: I liked the refocus of the paper to be a survey paper on Kalah rather than an algorithmic focus. It seems that the authors took the time and effort to address all the comments in a satisfactory way.
Response: Thank you! Your positive comments are much appreciated.
2) Comment: The abstract is not focused and the last couple of sentences does not make sense and do not connect well to the first part logically (e.g. the last two sentences do not make sense).
I suggest rewriting the abstract.
Response: Thank you for pointing out this issue. We have rewritten Abstract accordingly.
3) Comment: My comment on the statistical results is still standing, even with the new changes.
I do not think that 20 games will allow for a statistically significant difference between the algorithms. Anyway, regardless of my own intuition, a statistical test should be applied. The win rate difference when only 20 games were played is not enough, especially when apparent differences are so small.
Response: This is a pertinent and serious comment. We have asked our colleague, Dr. Martina Litschmannová, who is an expert in statistical methods, to help us with a suitable test (We have added her as another co-author of the manuscript). Since the data (i.e., the scores of both the players for every single game) are dependent (paired) giving the same sum of eventually captured seeds, the differences of scores can be taken for a statistical test. That is, a one-sample test can be performed instead of a two-sample test. We avoided testing apparent cases (e.g.,
Algorithm 2 with the tree-depth of 6 vs. the human player or vs. Algorithm 1). As first, it was verified whether the data sets are normally distributed by using the Shapiro-Wilk test. This test is limited to 3 to 30 samples; therefore, it fits well for our 20 games. Since some of the data were not normally distributed, we used nonparametric methods – the range estimation of score differences medians and the Wilcoxon two-tailed test for paired samples instead of the t-test.
The Wilcoxon test estimates the null value of the median of the differences. Again, this test is suitable for small sets (approx. up to 50 samples). The obtained interesting results for the firstset of tests are summarized in Table 3. The results made by the NOVA test for selected competing algorithms in the robin-round tournament are given in Figure 13.
This manuscript is a resubmission of an earlier submission. The following is a list of the peer review reports and author responses from that submission.
Round 1
Reviewer 1 Report
The paper presents some heuristic algorithms to play the Kalah
game. The game is a solved game (the first player wins) so the
results only have practical significance. The winning
percentage performance of the algorithms agree with the known
theoretical results. The performance of the algorithms shown in the
experimental evaluations are similar to a human player in the
scale of seconds, not too bad or too good considering the experiments
are performed on an ordinary laptop or desktop computer.
The paper has numerous grammar issues. Although in many cases
they do not greatly affect a reader's understanding, they had
better be cleared to make the paper more professional. Also, it
is my opinion that this paper should be shortened to at most
16 pages due to a lot of redundancy or over-detailed descriptions and nearly all sections need such improvement.
Reviewer 2 Report
The paper delves into the (much researched) game of Kalah and presents yet another heuristic function for the game. The heuristic is evaluated with an average human player and minimax algorithms using different heuristics. Overall, the paper is very well written. The breadth of the related work section on the history of the game, its many variations and the endless solutions and algorithms that were offered over the years is great and results in a very interesting piece to read. However, the paper has three main weakness: the first (and most important) is in its significance. I simply fail to capture the importance of yet another (ad-hoc) heuristic function for the game. I could not see any general scientific insight that can be used in any future research. Second, there are no theoretical contributions in the paper. Is it possible to say anything profoundly interesting (theoretically wise) about the proposed function\algorithm? Lastly, the experimental section is also weak as no statistical analysis was made to evaluate the significance of the results.
Here are some more comments I gathered along the reading:
- Page 2 – not all board games are zero-sum. For example, today there are cooperative board games (e.g. Pandemic) and other forms of games.
- Page 2 - last paragraph, I don’t agree with the claim that game-tree search require advanced programming effort and skills. Regarding [38], I think that 39GB of end-game database is not very complex to store these days.
- Page 3 – why run minimax without alpha-beta pruning (b.t.w change alfa-beta to alpha-beta in the text)? And why limit to depth level of 4? This is not reasonable path to take in the experiment as any simple laptop can easily be played to depth 8 in Kalha. Also, I advise sticking to even plys. I.e, search to levels 2, 4, 6 and 8.
- I really loved Section 2 of the paper
- Page 6 – I don’t understand what is ”advanced heuristic minimax”
- Page 8 – I do not think that the claim in line 322 is true. The related work section did see work that looks at the opponent modeling part and are suitable to real-time game.
- Page 8 – line 355 “by playing against can serve” ? cannot parse the sentence.
- Page 8 – lines 335-339 – this paragraph is very important for the significance of the work, but it is left weak without arguments towards the claims. I think this paragraph should be extended in future versions.
- Page 8 – 341 – not sure about the Chess branching factor. I know that it is around 35 moves (on average), how can Kalah be even remotely close to Chess’s branching factor?
- The experiment section is very weak. The depths, the comparison, the lack of statistical measures. It should be redone with more procedural thought into the process.
To sum up, the strengths and weaknesses of the work were presented above. I also do not think that the topic of the work is relevant to the “informatics” journal’s aim and scope. In light of the above, I want to advise the authors to reframe their work as a review work of the Kalah game, with some (minor) contribution in terms of the heuristic function. Such work would be much better fitted to journals like “AI review” or “IEEE transactions on games”.
Reviewer 3 Report
This is an interesting contribution to the literature of Kalah games. The over view is nice, and the human-like play algorithms are interesting. Apart from correcting small typo mistakes, such as "close to thinking of a real the real human player" in Section 8, I recommend this manuscript for publishing.